# Drivers of the Ectoparasite Community and Co-Infection Patterns in Rural and Urban Burrowing Owls

**DOI:** 10.3390/biology11081141

**Published:** 2022-07-29

**Authors:** Ángeles Sáez-Ventura, Antonio J. López-Montoya, Álvaro Luna, Pedro Romero-Vidal, Antonio Palma, José L. Tella, Martina Carrete, Gracia M. Liébanas, Jesús M. Pérez

**Affiliations:** 1Department of Animal and Plant Biology and Ecology, Jaén University, Campus Las Lagunillas, s.n., 23071 Jaén, Spain; asaezvent@gmail.com (Á.S.-V.); gtorres@ujaen.es (G.M.L.); 2Department of Statistics and Operational Research, Jaén University, Campus Las Lagunillas, s.n., 23071 Jaén, Spain; amontoya@ujaen.es; 3Estación Biológica de Doñana (CSIC), Av. Américo Vespucio, s.n., 41092 Sevilla, Spain; alvaro.luna@universidadeuropea.es (Á.L.); promvid1@upo.es (P.R.-V.); palmag.antonio@ebd.csic.es (A.P.); tella@ebd.csic.es (J.L.T.); 4Department of Health Sciences, Faculty of Biomedical and Health Sciences, Universidad Europea de Madrid, 28670 Madrid, Spain; 5Department of Physical, Chemical and Natural Systems, Pablo Olavide University, Ctra. de Utrera, km 1, 41013 Sevilla, Spain; mcarrete@upo.es; 6Wildlife Ecology & Health Group (WE&H), Universitat Autònoma de Barcelona, Bellaterra, 08193 Barcelona, Spain

**Keywords:** *Athene cunicularia*, co-infection, ectoparasites, fleas, lice, mites, urban ecology

## Abstract

**Simple Summary:**

We analyzed the ectoparasite community of a monomorphic and non-social bird, the burrowing owl, Athene cunicularia, breeding in rural and urban habitats. Such community was composed by two lice, one mite and one flea species. Rural individuals had more fleas and less mites than urban ones. Adult birds harbored less ectoparasites than young ones and females harbored more lice than males. The presence of lice was positively related to the presence of fleas. On the contrary, the presence of mites was negatively related to the presence of fleas and lice. The study of parasite communities in urban and rural populations of the same species can shed light on how urban stressor factors impact the physiology of wildlife inhabiting cities and, therefore, the host-parasite relationships.

**Abstract:**

Urbanization creates new ecological conditions that can affect biodiversity at all levels, including the diversity and prevalence of parasites of species that may occupy these environments. However, few studies have compared bird–ectoparasite interactions between urban and rural individuals. Here, we analyze the ectoparasite community and co-infection patterns of urban and rural burrowing owls, *Athene cunicularia*, to assess the influence of host traits (i.e., sex, age, and weight), and environmental factors (i.e., number of conspecifics per nest, habitat type and aridity) on its composition. Ectoparasites of burrowing owls included two lice, one flea, and one mite. The overall prevalence for mites, lice and fleas was 1.75%, 8.76% and 3.50%, respectively. A clear pattern of co-infection was detected between mites and fleas and, to less extent, between mites and lice. Adult owls harbored fewer ectoparasites than nestlings, and adult females harbored more lice than males. Our results also show that mite and flea numbers were higher when more conspecifics cohabited the same burrow, while lice showed the opposite pattern. Rural individuals showed higher flea parasitism and lower mite parasitism than urban birds. Moreover, mite numbers were negatively correlated with aridity and host weight. Although the ectoparasitic load of burrowing owls appears to be influenced by individual age, sex, number of conspecifics per nest, and habitat characteristics, the pattern of co-infection found among ectoparasites could also be mediated by unexplored factors such as host immune response, which deserves further research.

## 1. Introduction

Emerging infectious diseases have motivated substantial advances in understanding the ecology of parasite communities [1,2]. Several studies show that parasite-host interactions can vary among species, landscapes, and habitats [3,4], depending on the extent and complexity of their distributions, and host features: e.g., body size, age, behavior [5], and environmental characteristics: e.g., conspecific density, climate [6,7,8]. Moreover, seasonal movements and dispersal of host species across the landscape may also contribute to changing their parasite communities, also increasing the chances of transmission to other species [9]. Because host and parasite diversities are strongly connected, changes in the landscape are also likely to modify parasite diversity, with implications for wildlife but also human health [10,11].

Individuals may harbor parasites belonging to different zoological taxa (“island ecosystems” [2,12]). Although direct interactions among parasites are important in determining which combination of parasites will be present in an individual, e.g., mechanical facilitation [13], interference competition [14], indirect interactions (i.e., those mediated by host defenses [4]) are critical in explaining intracommunity variability. Associations between parasites may be similarly influenced by host traits such as behavior, ecology, exposure history, and pathologies [15], which may influence the transmission [16,17], distribution patterns [18], and the load patterns of parasites [19]. Moreover, morbidity induced by one parasite can affect host exposure to others, even if they are antagonistic [20], and mortality induced by one parasite could also reduce available hosts for other parasite species [21].

Urbanization constitutes a primary driver of habitat loss and fragmentation, affecting the spatial distribution of species and inducing deep ecological changes that affect many components of biodiversity [22,23,24]. Although animal communities are often simplified and homogenized in cities [25], these novel environments can also host a large number of species, including endangered ones [26,27,28], which may flourish and reach higher densities than in natural areas [27,29,30]. However, changes in conspecific density, resource availability and predation pressure commonly found in urban ecosystems [27,31,32,33] can have both positive and negative effects on the breeding performance and survival prospects of individuals [34,35,36]. Urbanization may also affect the parasite-host interactions by providing some effective resources to deal with ectoparasites: e.g., using cigarette butts within the nest material reduce the ectoparasite load in birds [37], or by increasing stress [38]: e.g., heavy metals [39], noise [40], human disturbance [41], but see [35], which can affect the body condition and health status of the individuals [42]. Thus, urban individuals can differ in their propensity to be parasitized compared to their rural counterparts.

Although our knowledge of changes in ecological and evolutionary processes associated with urban life has increased during recent years, few studies have explored differences in parasite-host interactions between conspecifics living in rural and urban habitats [43,44]. Here, we investigate the ectoparasite community of burrowing owls *Athene cunicularia*, a non-social, monogamous species [45] studied intensively as a model for understanding the drivers and consequences of urban invasion. The selection of this cavity-nester raptor is relevant as nesting cavities provide a stable microclimatic context for the development of ectoparasites that feed on avian hosts [46]. Moreover, because burrowing owls may share their burrows with mammals over time (e.g., armadillos Dasypodidae, skunks *Conepatus* sp., or vizcachas *Lagostomus maximus* [47]), they are likely to host mammalian ectoparasites and, thus, a wider variety of ectoparasites than other avian species. Previous research conducted in the same owl population has shown that densities are notably higher in the city [27], urban individuals being more philopatric and productive than rural ones [27,35,48]. Rural owls breed in natural grasslands and pastures dedicated to cattle and crops, where human presence is rare and restricted to some scattered roads. Urban owls, conversely, breed in private gardens, public parks, free spaces plots among houses, roundabouts, and large avenues, in continuous contact with people and traffic [35]. The diet of the burrowing owl mainly comprises arthropods and small mammals, but also small birds, reptiles, and amphibians [49,50], without differences among urban and rural breeding territories [51]. During the breeding season, prey are usually consumed inside or at the entrance of the burrows, which are used not only for breeding but also as a refuge [52], increasing the likelihood of contact between owls, prey and potential parasites. Ectoparasite prevalence and load would be higher in nestlings, which need to mature the immune response, particularly, complement activation [53], than in adults, and in nests with more conspecifics, which are more frequent in urban than rural areas [54]. Besides, owls in nests located in more arid areas would host less ectoparasites due to the negative influence of aridity on arthropod survival [55,56]. Testosterone has been linked to increased parasite susceptibility in different vertebrate groups, whereas estrogen is often associated with increased resistance against infection [56,57], so it is expected that males to harbor more ectoparasites than females. As nutritional resources can govern both growth and resistance to ectoparasites [58,59], a negative relation between owl body mass and ectoparasite load can be expected. Finally, differences in ectoparasite load between urban and rural individuals associated with differences in conspecific density but also as a consequence of the relative absence of other wild animals using burrows in cities can also be expected. Specifically, we analyzed (i) the influence of individual traits (e.g., sex, age, body mass), and environmental conditions (e.g., number of conspecifics per burrow, habitat type, and the aridity of the area) on ectoparasite prevalence, and (ii) the patterns of ectoparasites co-infection.

## 2. Materials and Methods

### 2.1. Study Area and Species

Our study area encompasses 5400 km^2^, including the urban area of Bahia Blanca and its surrounding rural expanses (province of Buenos Aires, Argentina) (Figure 1). There, we monitored the breeding populations of burrowing owls from 2006 to 2020 [36]. Urban nests were located in private and public gardens, vacant lots among houses, curbs of the streets, roundabouts, and large avenues, in contact with the intense daily activity derived from cities. Rural nests, on the contrary, were located in large extensions of natural or semi-natural grasslands, with very low human presence. It is worth noting that the city is immediately surrounded by large areas of pastures, and there is no obstacle precluding the movement of individuals between urban and rural areas. Moreover, as these owls are able to excavate their own burrows, their distribution is not constrained by the availability of nesting structures [35]. Studies focused on this host species have reported more than 60 parasite species across its distribution range, including one virus, seven protists, seven trematodes, six nematodes, one acanthocephalan, 14 acari (including soft ticks, Argasidae, and mites), three hippoboscid diptera, one carnid fly, 25 fleas and two lice (Appendix A).

### 2.2. Sample Collection

During the breeding periods (from November to early February) of 2016–2017 and 2017–2018, we captured 482 urban and 387 rural burrowing owls occupying 424 nests (226 urban and 198 rural nests; Figure 1) using bow nets and ribbon carpets. Birds were marked with an individually numbered plastic color ring readable at distance, aged as chick (including fledgling) or adult, inspected for ectoparasites, measured (wing length, in mm), and weighed (in g) before releasing them. The number of individuals occupying each nest, i.e., breeders, nestlings, and individuals delaying dispersal [48], was obtained by intense monitoring throughout the breeding season [36,45]. The sex of adults was assigned by plumage pattern coloration [60] or, when needed for adults and nestlings, by molecular procedures using blood samples [42].

Ectoparasites were searched for on all parts of the owl’s body by two people during a 5 min inspection, pulling the feathers apart to detect individuals among them and on the skin, with special emphasis on the wings, head, and trunk. All ectoparasites found were removed, counted, and fixed in Eppendorf tubes filled with 97% ethanol for latter identification in the lab. Ectoparasites were treated and mounted in Canada balsam [61] to be identified based on published descriptions: lice [62,63,64,65]; fleas [66,67,68]; mites: [69].

### 2.3. Statistical Analysis

We calculated the prevalence of each ectoparasite species according to [70,71] and estimated their abundances as the number of each ectoparasite (lice, fleas, and mites) removed from a bird during 5 min. We used generalized linear latent variable models (GLLVMs) to assess factors affecting the ectoparasite community composition, considering patterns of species co-infection [72,73]. GLLVMs were fitted using the gllvm package [74] in R (version 3.3.2) [75], which incorporates latent variables derived from the Laplace approximation method implemented through Template Model Builder [76] to investigate species co-infection patterns while considering the effects of explanatory variables. GLLVMs combine separate species Generalized Linear Models (GLMs) with the effect of explanatory and latent variables, which account for any residual covariation explained by unknown variables or factors not included in the model [77]. GLLVMs also allowed us to separate the relative effect of explanatory and latent variables on ectoparasite co-infection patterns (due to the extremely low prevalence of *Strigiphilus speotyti*, the two lice species were grouped into “lice” for this analysis) and to estimate the strength and direction (+/−) of the correlations. We fitted a GLLVM considering ectoparasite abundance as dependent variable (zero-inflated Poisson distribution, log link function). The model included individual body mass, sex, age (nestling or adult), number of individuals per nest, habitat and aridity as explanatory variables, plus the interaction between age and sex. Aridity was calculated as the Gaussen aridity index GI = ∑ mean precipitation—(2 × mean temperature) [78]. Data on daily temperature and precipitation for the days employed for samples collection were obtained from a local meteorological station. We also included the year as a fixed factor and, to avoid pseudoreplication, the nest as random term in the model. We checked the strength and sign of correlations between species co-infection using the 95% confidence interval. The mean coefficients for each variable (and their 95% confidence intervals) were plotted to determine which coefficients corresponding to the explanatory variables were statistically significant. The goodness of fit of the model was checked graphically using Dunn–Smyth residuals [79] and a normal-quantile plot of the residuals using the summary function of the gllvm package.

## 3. Results

### 3.1. Ectoparasite Community

We collected four ectoparasite species in our burrowing owl populations: one mite belonging to the family Laelapidae (order Mesostigmata) (*n* = 39), two chewing lice (*Strigiphilus speotyti* Ischnocera: Philopteridae (*n* = 2), and *Colpocephalum pectinatum* Amblycera: Menoponidae) (*n* = 224), and one flea (*Polygenis platensis* Rhopalopsyllidae) (*n* = 51). With the samples obtained, we were unable to identify the mite to species level (for more details, see Appendix A). Only 106 of the birds sampled (12.2%) harbored any ectoparasite: 76 (8.76%) had lice (8.64% *C. pectinatum*, 0.23% *S. speotyti*, and 0.12% had both species), 30 (3.50%) had fleas, and 15 (1.73%) were parasitized by mites. Moreover, 4.7% of owls simultaneously had lice and fleas, 5.6% were parasitized by lice and mites, 3.7% by fleas and mites, and 0.9% had all three parasite types. No bird harbored all the four ectoparasites simultaneously (Table 1).

We found a lower prevalence of mites, fleas and lice in 2017–2018 than in 2016–2017. Adult owls had fewer mites, lice, and fleas than nestlings, and adult females harbored more mites than males. Individuals occupying burrows in rural areas had more fleas and fewer mites than those inhabiting urban areas. Nevertheless, habitat did not influence the number of lice. The number of mites and fleas was correlated positively with the number of owls in the family unit, but lice showed the opposite pattern. The number of mites was negatively related to aridity (GI), while the number of fleas increased along this gradient. Body mass did not influence the lice number, but affected negatively both fleas and mites abundance (Figure 2; Table 2).

### 3.2. Co-Infection Pattern

The correlated response model (CRM) including explanatory variables shows a significant positive correlation between lice and fleas and a negative correlation between mites and fleas and mites and lice (Figure 3A). However, in the CRM considering latent variables, only positive correlation between lice and fleas remains significant (Figure 3B).

## 4. Discussion

### 4.1. Ectoparasite Community and Co-Infection Pattern

In recent years, many studies have contributed to a better understanding of the ectoparasites host by burrowing owls through their distribution range. In this study, using a notably larger sample size than in the rest of the studies [80,81], we found that approximately 12% of the individuals examined in two consecutive breeding seasons harbor at least one of four ectoparasite species. The flea species detected, *Polygenis platensis*, has been previously collected from mammals and birds from South American countries, including burrowing owls in Argentina [68,82]. This species is a common parasite of rodents and insectivorous mammals, but also infests carnivores (Procyonidae and Canidae) and humans [67,83]. The presence of this flea is relevant because Siphonaptera insects are more associated with mammals than birds, and cases of fleas shared between mammals and raptors are not common [83]. In our case, the presence of *P. platensis* can be explained by its presence in owl’s preys and by the atypical habit of owls to use subterranean holes for nesting and resting, sometimes breeding in burrows previously used by mammals [36,47,48], which provides a context for further cross-species parasitization. Inside the burrows, the temperature is more constant and the relative humidity is much higher than outside [84]. This fact, together with the accumulation of decaying material (burrowing owls decorate the entrance of their burrows with different organic materials [85]) may favor the survival of arthropods in general, and ectoparasites in particular, so that fleas can be even more abundant and prevalent in the burrows than in the individuals themselves [81]. As for the lice recorded, the same species have been identified in burrowing owls in their Northern and Southern range: *Strigiphilus speotyti* has been found in individuals from the USA, Argentina, Chile and Brazil, while *Colpocephalum pectinatum* appeared in studies performed in the USA, Argentina, Brazil and Mexico (see details in the Appendix A). *Colpochepalum* species have been detected in more than ten orders of birds, including owls [62,86], but *Strigiphilus* are exclusive to the order Strigiformes [64,86,87]. Fleas, can be carried by the rodents preyed on by owls, which are often left at the entrance or inside the nest to feed nestlings. Furthermore, burrowing owls not only serve as potential sources of food, but can also be used as phoretic hosts, giving protection to such arthropods from predators, providing thermoregulatory advantages, and facilitating access to habitat for larvae development within the nest substrate [59]. Finally, mites from the Laelapidae family found in our burrowing owl populations include several species both free-living and others that live in association with a wide variety of vertebrates and invertebrates [88,89], so the presence of an unidentified species in our sampled birds is not surprising. Further studies are needed to identify the mite species involved and to understand the nature of the mite-owl interaction.

Host defense (preening) mediates interspecific competition [90]. Abundances of lice and fleas were positively correlated suggesting that facilitation processes may occur, probably mediated by host immune response [91]. The contrary happens regarding mites and fleas, and mites and lice, and could be explained by both competition between such ectoparasite groups and/or their negative effect on feather development [92]. On the other hand, the status of host feather molt and its impact on ectoparasite load [92] remain unknown. Some feather mites show a mutualistic relationship with their hosts (e.g., cleaning host feathers [93]. To properly explain the relationship between mites and the other ectoparasite groups it is necessary to identify the mite species involved and its diet as well, to elucidate the true nature of this symbiotic relationship.

### 4.2. Individual and Environmental Factors Explaining Infection Pattern

No lesions nor disease symptoms were detected in the sampled birds. Our results show that adult owls had fewer ectoparasites (mites, fleas, and lice) than nestlings, coinciding with other studies [94,95,96]. Allopreening (mutual preening), whereby the bill is used to preen the partner’s feathers, is relatively common in burrowing owls [97,98], and more frequent among adults. This behavior, which has social functions such as reducing stress levels [99,100], reinforcing pair bonds [101], and reinforcing social hierarchies [102,103], could explain the lower prevalence of ectoparasites in adults compared with nestlings. In addition, the permanence of a large number of nestlings inside the nests (we observed nests with up to 5–7 fledglings) during their first weeks of life may favor parasitization and parasite exchange between siblings. Further research is needed to understand whether these results may be also due to the lower immunocompetence of nestlings, or to the better nutritional state of adults, or if it is related to other environment or behavioral factors not considered here [42].

Male and female burrowing owls have similar ectoparasite burden, regarding fleas and lice, which is common in monomorphic birds [104], but females harbored more mites (Figure 2). During incubation and the first weeks of chick rearing, females spend more time inside the nest or at its entrance, being potentially more exposed to parasites. As before, the differences observed in mite load may be also due to differences in the immune response [105,106], which could differ not only among individuals but also between males and females, mainly during the breeding period [107].

Body mass is a reliable indicator of offspring survival in birds and mammals [108]. It is assumed that heavy birds are able to produce an immune response while maintaining a relatively large uropygial gland [106]. In *Athene cunicularia*, the uropygial gland accounts for ca. 9% of its weight [109]. This could explain the negative relationship between flea and mite burden and owl body mass. Amblyceran lice (as is the case of *C. pectinatum*) occur in contact with and fed in host skin, even obtaining blood. So, it is expected that amblyceran lice abundance and richness have evolved in response to interaction with the immune system of the host [110]. In our case, owl body mass did not affect the number of lice (*C. pectinatum*). This suggests that the immune response to this amblyceran louse could be very mild or moderate.

Few studies have compared the ectoparasite burden of urban and rural conspecifics. For instance, Ancillotto et al., 2018 [111] have found a positive relationship between the degree of urbanization and the prevalence and abundance of chewing lice and mites in exotic parakeets in Italy. Wemer et al., 2021 [38] found fewer ectoparasites, lower haemolysis and lower body mass index in nestling Eurasian kestrels (*Falco tinnunculus*) from more urbanized areas compared to those inhabiting less urbanized areas. In our study, we found differences in the abundance of fleas and mites among owls nesting in urban and rural areas. Although urban and rural owls include rodents in their diet in similar proportions [58,112], the presence of mammals that may also breed or shelter in their burrows may explain why rural owls acquire more fleas than urban ones. Differences in burrow microclimate can also influence ectoparasite distribution [113].

Flea numbers increased with increasing GI values (that means high rainfall values and/or low temperature), since such conditions favor arthropod survival [51]. The immature stages of fleas developing in host burrows are sensitive to air temperature and humidity, with effects on both development and survival times [114]. Larval survival may be compromised under low humidity conditions (e.g., 40–50%) [115]. On the other hand, increasing humidity of great tit (*Parus major*) cavity-nests as a result of flea infestations has been reported [116].

## 5. Conclusions

Today, as global human population continues to increase and urbanize, scientists, conservationists, and politicians agree that understanding the patterns that explain the biodiversity of cities is a priority for urban planning and nature conservation [117]. Additionally, there is a growing concern towards the interaction between wildlife diseases and urban habitats, due to their implications for wildlife conservation and public health. With this study, integrated in a long-term monitoring program of burrowing owls, we contribute to understanding the infestation patterns of a raptor species inhabiting both rural and urban areas, showing how individual and environmental factors influence the ectoparasite loads and the co-infection patterns. Further research is needed to explore the influence of the immune system in the effects caused to the host by the presence and activity of the ectoparasites detected. A more detailed exploration, including also more host species, could help to understand the cross-infection between individuals and even between species. Seasonal characterization of reproductive hormones and stress levels of burrowing owls could explain the patterns observed in this study. Equally, extending our study to endoparasites could offer a complete interpretation of the interactions between parasites and burrowing owls under different scenarios. Finally, the analysis of the potential impact of parasites in the breeding performance and survival prospects of individuals inhabiting the city and rural areas could contribute to acquiring a more complete understanding of the recent colonization of the city by burrowing owls.

## Figures and Tables

**Figure 1 biology-11-01141-f001:**
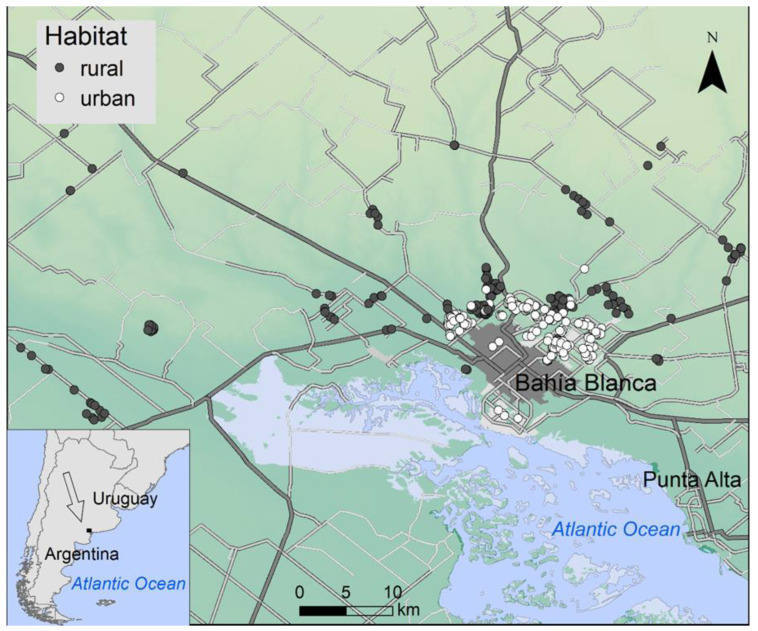
Location of the study area. Grey and white dots show the location of rural and urban nests sampled, respectively. Paved (grey) and unpaved (white) roads are also shown.

**Figure 2 biology-11-01141-f002:**
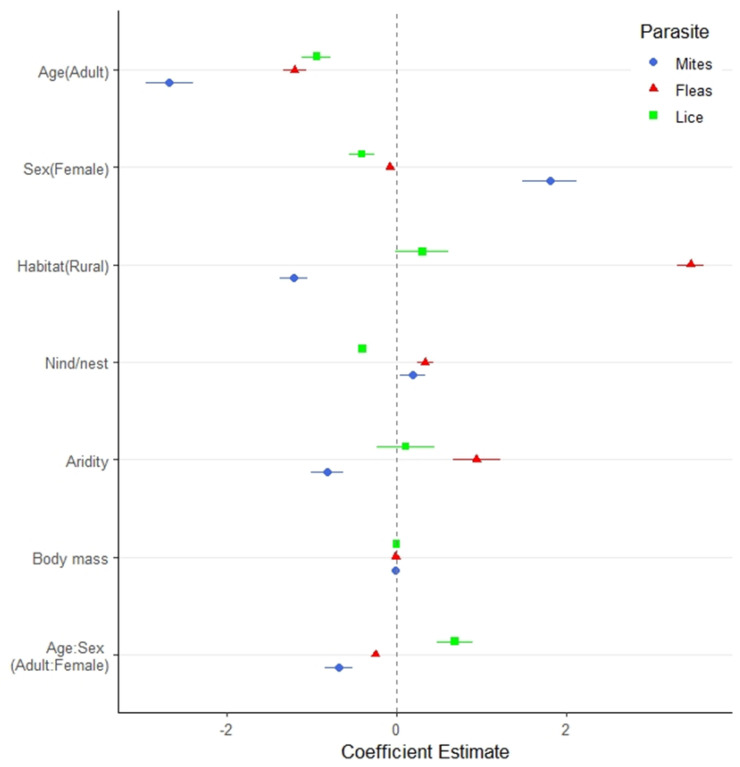
Influence of individual traits (sex, age, body mass), and environmental conditions (number of conspecifics per burrow, habitat type, and the aridity of the area) on the abundance of mites, fleas and lice in the burrowing owls *Athene cunicularia*. Estimated coefficients (and 95% confidence intervals) of the covariates for the three ectoparasite groups. 95% confidence intervals overlapping 0 (dashed vertical line) indicate a non-significant coefficient.

**Figure 3 biology-11-01141-f003:**
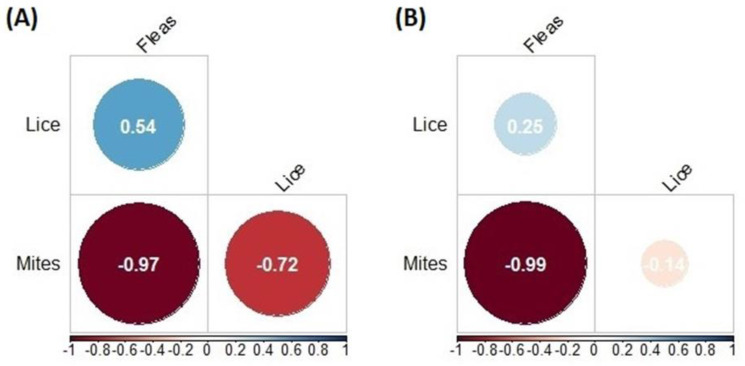
Correlations between abundance of ectoparasites due to explanatory (**A**) and latent (**B**) variables based on the CRM. Significant correlations (i.e., 95% confidence interval not overlapping zero) are represented by solid colours, while transparent ones represent non-significant correlations. The strength of correlations is represented by the size of the circles.

**Table 1 biology-11-01141-t001:** Prevalence and abundance of ectoparasites found in rural and urban burrowing owls *Athene cunicularia* in Bahía Blanca (Argentina). SD: standard deviation; SU: sex undetermined.

Burrowing Owls		Mites			Lice			Fleas		
	n	Prevalance	Mean ± SD	Range	Prevalence	Mean ± SD	Range	Prevalence	Mean ± SD	Range
Global	869	1.75%	2.60 ± 4.31	1–18	8.76%	3.01 ± 3.12	1–17	3.50%	1.70 ± 1.02	1–5
Males	237	0.70%	4.00 ± 6.87	1–18	4.21%	3.69 ± 4.01	1–17	1.52%	1.77 ± 1.24	1–5
Females	301	1.05%	1.67 ± 0.71	1–3	4.56%	2.38 ± 1.82	1–8	1.99%	1.65 ± 0.86	1–3
SU	331	1.61%	1.00 ± 0.00	1–1	4.94%	2.60 ± 2.32	1–9	2.30%	1.95 ± 1.13	1–5
Chicks	380	1.52%	2.69 ± 4.63	1–8	5.02%	3.37 ± 3.59	1–17	2.22%	1.95 ± 1.13	1–5
Adults	489	0.23%	2.00 ± 1.41	1–3	3.74%	2.53 ± 2.32	1–9	1.29%	1.27 ± 0.65	1–3
Rural	387	0.35%	1.67 ± 1.15	1–3	4.09%	3.43 ± 3.97	1–17	2.69%	1.65 ± 1.07	1–5
Urban	482	1.40%	2.83 ± 4.80	1–18	4.67%	2.65 ± 2.11	1–8	0.82%	1.86 ± 0.90	1–3
2016–2017	413	3.39%	2.71 ± 4.44	1–18	9.20%	3.26 ± 3.87	1–17	3.87%	1.81 ± 1.16	1–5
2017–2018	456	2.19%	1.00 ± 0.00	1–1	7.89%	2.58 ± 1.87	1–8	2.41%	1.72 ± 0.90	1–3

**Table 2 biology-11-01141-t002:** Results of the estimated coefficients of the gllvm model.

Parasite	Component	Estimate	Std. Err.	z Value	*p*-Value
Lice	Age	−0.9389	0.0879	−10.682	<2 × 10^−16^ ***
	Sex	−0.4041	0.0745	−5.426	5.76 × 10^−8^ ***
	Habitat	0.3051	0.1598	1.909	0.0562 *
	Nind/nest	−0.3988	0.0022	−179.647	<2 × 10^−16^ ***
	Aridity	0.1138	0.1752	0.650	0.51577
	Body mass	0.0004	0.0007	0.603	0.54620
	Age:Sex	0.6932	0.1091	6.356	2.07 × 10^−10^ ***
Fleas	Age	−1.1969	0.0683	−17.514	<2 × 10^−16^ ***
	Sex	−0.0727	0.0125	−5.818	5.94 × 10^−9^ ***
	Habitat	3.4717	0.0805	43.115	<2 × 10^−16^ ***
	Nind/nest	0.3418	0.0469	7.287	3.17 × 10^−13^ ***
	Aridity	0.9466	0.1409	6.718	1.84 × 10^−11^ ***
	Body mass	−0.0088	0.0008	−11.187	<2 × 10^−16^ ***
	Age:Sex	−0.2483	0.0045	−54.809	<2 × 10^−16^ ***
Mites	Age	−2.6764	0.1419	−18.862	<2 × 10^−16^ ***
	Sex	1.8086	0.1631	11.090	<2 × 10^−16^ ***
	Habitat	−1.2081	0.0805	−15.003	<2 × 10^−16^ ***
	Nind/nest	0.1976	0.0761	2.593	0.00951 **
	Aridity	−0.8150	0.0948	−8.600	<2 × 10^−16^ ***
	Body mass	−0.0103	0.0010	−9.963	<2 × 10^−16^ ***
	Age:Sex	−0.6797	0.0830	−8.190	2.62 × 10^−16^ ***

*** *p*-value < 0.001; ** *p*-value < 0.01; * *p*-value < 0.05.

## Data Availability

Data are available from the authors upon reasonable request.

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
