# Peer review of "Drivers of the Ectoparasite Community and Co-Infection Patterns in Rural and Urban Burrowing Owls"

_biology, 2022, doi:10.3390/biology11081141_

Round 1

Reviewer 1 Report

This journal article “Drivers of the ectoparasite community and co-infection patterns in rural and urban burrowing owls” gives a detailed insight into the ectoparasite communities in borrowing owls. The main objective is to identify the species of ectoparasites and then evaluate if there is any correlation between different variables, both individual and environmental.
I found this article really interesting, and the number of sampled animals is consistent, underlining a consistent field work.
Thus, I consider your article ok for publication but after some modifications, to improve its quality.

Simple summary:

Delete “at”. breeding in both at rural and urban habitats

Abstract:

I suggest you to add (at least) the general prevalence of mice, flea and lice hosted on owls

the pattern of co-infection found among ectoparasites could also be mediated by

Introduction:

Line 50. Rephrase suggestions.
of their distributions, and host features (e.g., body size, age, behavior [5]) and environmental (e.g., conspecific density, climate [6-8]) characteristics.

Line 48-51 and 57-61. If possible, In these parts I suggest to remove the brackets to try to make the sentences more fluent. If the sentence is too long, split it in two parts.

Line 61-64. Suggestion: because the word “parasite” is repeated 4 times in the same sentence. I suggest to rephrase it. As an example: “which may influence then the transmission [19,20], distribution patterns [21], and the load of parasites [22].” Or something similar

Line 71: these novel environments can also host a large number of spe-

Line 76-80: as for the previous lines, I suggest to remove the brackets, and try to make the text more fluid.

Line 77-78. I am impressed and curious about this finding about cigarette butts. I checked the reference (41) and it seems that there are no info about cigarette butts in the paper. May you check if the reference numbers and papers match correctly?

Lines 92-96. I suggest to move these “objective“ lines at the end of the introduction.

“Specifically, we analyzed (i) the influence of individual traits (e.g., sex, age, body mass), and environmental conditions (e.g., number of conspecifics per burrow, habitat type, and the aridity of the area) on ectoparasite prevalence, and (ii) the patterns of ectoparasites coinfection”

Line 98-110. In this part the contents are good, but have to be reshaped. I kindly suggest not to use the personal form “we” but try to describe the scientific findings in a neutral and impersonal way.

Materials and methods

Line 115-128. This part is good in content but I suggest you to move it in the introduction part, as it describes the biology of the species and the previous researches that have been conducted on this topic. This is not part of your study methods.

Line 152. during a 5 minute inspection.

Line 139-140. I suggest to remove some references. 5 inside the same sentence, to support a single concept, are too much.

Results

Line 184-188. I would suggest you to explicit, near the prevalences that you describe, also the real numbers of parasitized animals, to have a better overview of the quantities. At least for the main points.   

Then, I would suggest also to explicit the total number of ectoparasites collected. How many are they? How many lice, fleas and mites?

Table 1.
Comment 1. In the Material and methods section you say that you captured 413 owls from urban and 456 from rural areas. In the table, the numbers do not match! I see a total of 482 in urban and 387 in rural. Can you kindly check which are the correct results?

Comment 2. The table is mainly about the ectoparasites identified in your study. Thus, I would highlight more clearly that the first two columns are about owls, because at a first sight it can be confusing.

Discussion.

Some comments/suggestions. You described that the animal weight was only related to the mites abundance. But then, which was their weight?
in the Materials and Methods section you say that you measured the wing length (mm) and weight (grams) but these data are not provided in the Results. Why?
I think that, if described in the MandM part, you should spend a brief description/comments about them also in the following sections.

in addition, studied animals were in a good BCS? Did you notice lesions or any noticeable record about their health status? I kindly suggest you to spend few words about it, to make the animal sampling/inspection more accurate.

Lina 244-246. I found this sentence contradictory. Before you say “but Strigiphilus are exclusive to the order Strigiformes [67,89,90].” And then you add that “Lice that parasitize owls, but also fleas, can be carried by the rodents preyed by owls”. But if these parasite species are specific of owls, you should not find them on rodents? I kindly ask you if you can better explain this part. 

Author Response

Reviewer 1

This journal article “Drivers of the ectoparasite community and co-infection patterns in rural and urban burrowing owls” gives a detailed insight into the ectoparasite communities in borrowing owls. The main objective is to identify the species of ectoparasites and then evaluate if there is any correlation between different variables, both individual and environmental.

I found this article really interesting, and the number of sampled animals is consistent, underlining a consistent field work.

Thus, I consider your article ok for publication but after some modifications, to improve its quality.

Authors response: thank you very much

Simple summary:

Delete “at”. breeding in both at rural and urban habitats

Authors response: done

Abstract:

I suggest you to add (at least) the general prevalence of mice, flea and lice hosted on owls

Authors response: done

the pattern of co-infection found among ectoparasites could also be mediated by

Authors response: done

Introduction:

Line 50. Rephrase suggestions.

of their distributions, and host features (e.g., body size, age, behavior [5]) and environmental (e.g., conspecific density, climate [6-8]) characteristics.

Authors response: done

Line 48-51 and 57-61. If possible, In these parts I suggest to remove the brackets to try to make the sentences more fluent. If the sentence is too long, split it in two parts.

Authors response: the brackets have been removed as suggested.

Line 61-64. Suggestion: because the word “parasite” is repeated 4 times in the same sentence. I suggest to rephrase it. As an example: “which may influence then the transmission [19,20], distribution patterns [21], and the load of parasites [22].” Or something similar

Authors response: done.

Line 71: these novel environments can also host a large number of spe-

Authors response: done

Line 76-80: as for the previous lines, I suggest to remove the brackets, and try to make the text more fluid.

Authors response: done

Line 77-78. I am impressed and curious about this finding about cigarette butts. I checked the reference (41) and it seems that there are no info about cigarette butts in the paper. May you check if the reference numbers and papers match correctly?

Authors response: we have checked them

Lines 92-96. I suggest to move these “objective“ lines at the end of the introduction.

“Specifically, we analyzed (i) the influence of individual traits (e.g., sex, age, body mass), and environmental conditions (e.g., number of conspecifics per burrow, habitat type, and the aridity of the area) on ectoparasite prevalence, and (ii) the patterns of ectoparasites coinfection”

Authors response: done

Line 98-110. In this part the contents are good, but have to be reshaped. I kindly suggest not to use the personal form “we” but try to describe the scientific findings in a neutral and impersonal way.

Authors response: done

Materials and methods

Line 115-128. This part is good in content but I suggest you to move it in the introduction part, as it describes the biology of the species and the previous researches that have been conducted on this topic. This is not part of your study methods.

Authors response: done

Line 152. during a 5 minute inspection.

Authors response: done

Line 139-140. I suggest to remove some references. 5 inside the same sentence, to support a single concept, are too much.

Authors response: done

Results

Line 184-188. I would suggest you to explicit, near the prevalences that you describe, also the real numbers of parasitized animals, to have a better overview of the quantities. At least for the main points.  

Authors response: done

Then, I would suggest also to explicit the total number of ectoparasites collected. How many are they? How many lice, fleas and mites?

Authors response: done

Table 1.

Comment 1. In the Material and methods section you say that you captured 413 owls from urban and 456 from rural areas. In the table, the numbers do not match! I see a total of 482 in urban and 387 in rural. Can you kindly check which are the correct results?

Authors response: it has been corrected.

Comment 2. The table is mainly about the ectoparasites identified in your study. Thus, I would highlight more clearly that the first two columns are about owls, because at a first sight it can be confusing.

Authors response: done

Discussion.

Some comments/suggestions. You described that the animal weight was only related to the mites abundance. But then, which was their weight?

in the Materials and Methods section you say that you measured the wing length (mm) and weight (grams) but these data are not provided in the Results. Why?

I think that, if described in the MandM part, you should spend a brief description/comments about them also in the following sections.

Authors response: we’ll try to answer simultaeously these 3 questions. In the material and methods section, we described in detail all the procedures carried out with each bird sampled. Since the biometry of Athene cunicularia has been previously addressed by a number of authors, we think this information is not relevant for this manuscript. Some data (e.g., wing length) can be used as a complementary criterium for bird sexing. Only bird weight was used here as explanatory variable, and it was correlated only with mite abundance. On the other hand, as weight values obatined fall within the range given by other authors. 

in addition, studied animals were in a good BCS? Did you notice lesions or any noticeable record about their health status? I kindly suggest you to spend few words about it, to make the animal sampling/inspection more accurate.

Authors response: done (see line 298 in the new version of the ms).

Line 244-246. I found this sentence contradictory. Before you say “but Strigiphilus are exclusive to the order Strigiformes [67,89,90].” And then you add that “Lice that parasitize owls, but also fleas, can be carried by the rodents preyed by owls”. But if these parasite species are specific of owls, you should not find them on rodents? I kindly ask you if you can better explain this part.

Authors response: you have reason; this sentence has been rephrased and we hope the current version is clear enough.

Reviewer 2 Report

The authors analysed the ectoparasite community of nesting burrowing owls. They found differences between urban and rural habitats, between adults and nestlings and between male and female adult birds. They also found patterns of co-infection between different ectoparasites.

The study is interesting and generally well written. I have few comments, see below.

Line 22: delete "both at"

Line 25-26: Delete "vice versa" in this sentence, it is unnecessary. You could also join this with the following sentence. 

Line 60: delete the ) after [16]

Line 105: add "to" between males and harbor

Line 107: please, correct the spelling of ectoparasite

Line 164-165: Here you talk about "a GLLVM", but in the next sentence you refer to "models". Then is not clear how the ectoparasite abundance variable was obtained. Is this the overall abundance (i.e. all ectoparasite species combined) or rather did you use multiple models for each ectoparasite species?  

Line 170-171: With only two years it might just be worth to compare the overall abundance of ectoparasites between years to see if there is any difference (which seems the case by looking at Table 1). This would allow to drop year from the GLLVM model to avoid over parameterization. Furthermore, the factor year is not reported in any figure or table in the text or supplementary material.  

Line 202: I think Table S1 is more informative than Table 1, and more interesting as part of your main question, i.e. what contributes to variation in the ectoparasite community. I would switch these two tables by presenting Table S1 in the main text and Table 1 in the supplementary material.

Line 210: please, define CRM

Line 212-213: I am confused. From the two plots of Figure 3 it seems to me that there is a strong negative correlation between mites and fleas in both cases. And that with latent variables the only positive correlation (lice and fleas) looses significance when compared with explanatory variables (transparent colour as you write in the legend of Figure 3)

Line 246: Do you mean that lice can parasitise fleas as well? Or rather than lice, as well as fleas, can be carried by rodents?

Line 256-257: How is immune response facilitating abundance of lice and fleas? Perhaps it is worth to explain this point in a clearer manner

Line 262: Change "for" with "to"

Line 264: Previously you wrote that you could not identify the mite species, so how do you know that this mite has a symbiotic relationship with the owls? I would just write that further studies are needed to understand the nature of the mite-owl interaction.

Line 281-282: Add "to" between exposed and parasites.

Line 288-289: This info is kind of lost here, as Figure 2 does not show this negative relationship. This would only be evident by presenting Table S1 in the main text instead of Table 1

Line 292: Change "on" with "in"

Lines 316-317: Change with "Additionally, there is a growing concern towards ....."

Line 320: change with "to the understanding"

Author Response

Reviewer 2

The authors analysed the ectoparasite community of nesting burrowing owls. They found differences between urban and rural habitats, between adults and nestlings and between male and female adult birds. They also found patterns of co-infection between different ectoparasites.

The study is interesting and generally well written. I have few comments, see below.

Authors response: thank you very much

Line 22: delete "both at"

Authors response: done

Line 25-26: Delete "vice versa" in this sentence, it is unnecessary. You could also join this with the following sentence.

Authors response: done

Line 60: delete the ) after [16]

Authors response: done

Line 105: add "to" between males and harbor

Authors response: done

Line 107: please, correct the spelling of ectoparasite

Authors response: done

Line 164-165: Here you talk about "a GLLVM", but in the next sentence you refer to "models". Then is not clear how the ectoparasite abundance variable was obtained. Is this the overall abundance (i.e. all ectoparasite species combined) or rather did you use multiple models for each ectoparasite species? 

Authors response: you have reason. The gllvm model is a multivariate model, where the dependent variable is multivariate (the three species of ectoparasites combined). To avoid confusion, we have changed “models” by “model”.

Line 170-171: With only two years it might just be worth to compare the overall abundance of ectoparasites between years to see if there is any difference (which seems the case by looking at Table 1). This would allow to drop year from the GLLVM model to avoid over parameterization. Furthermore, the factor year is not reported in any figure or table in the text or supplementary material. 

Authors response: despite at first sight, there seemed to be significant differences by years according to Table 1, the first gllvm model that was fitted did not detect significant differences, so we decided later to readjust the model without including the variable year, which is how it is now.

Line 202: I think Table S1 is more informative than Table 1, and more interesting as part of your main question, i.e. what contributes to variation in the ectoparasite community. I would switch these two tables by presenting Table S1 in the main text and Table 1 in the supplementary material.

Authors response: we partially agree with you. We have moved Table S1 to the main text (as Table 2), but we maintain Table 1 as it includes basic information about epidemiology (prevalence and intensity of parasitation) of ectoparasites studied.

Line 210: please, define CRM

Authors response: this means: correlated response model. It has been explained (see line 236 of the current version of the manuscript.

Line 212-213: I am confused. From the two plots of Figure 3 it seems to me that there is a strong negative correlation between mites and fleas in both cases. And that with latent variables the only positive correlation (lice and fleas) looses significance when compared with explanatory variables (transparent colour as you write in the legend of Figure 3)

Authors response: this is just what Figure 3 means. You have interpreted it correctly.

Line 246: Do you mean that lice can parasitise fleas as well? Or rather than lice, as well as fleas, can be carried by rodents?

Authors response: we mean that fleas can be carried by rodents, which can also use the burrow and/or they can be preyed by burrowing owls.

Line 256-257: How is immune response facilitating abundance of lice and fleas? Perhaps it is worth to explain this point in a clearer manner

Authors response: co-infection may present either as a pre-existing pathogen which is accentuated by the introduction of a new pathogen or may appear in the form of new infection acquired secondarily due to a compromised immune system.

Line 262: Change "for" with "to"

Authors response: done

Line 264: Previously you wrote that you could not identify the mite species, so how do you know that this mite has a symbiotic relationship with the owls? I would just write that further studies are needed to understand the nature of the mite-owl interaction.

Authors response: the sentence has been completed as you suggested, and we think it has gain clarity.

Line 281-282: Add "to" between exposed and parasites.

Authors response: done

Line 288-289: This info is kind of lost here, as Figure 2 does not show this negative relationship. This would only be evident by presenting Table S1 in the main text instead of Table 1

Authors response: we have moved Table S1 to the main text, as you suggested. Now it is Table 2.

Line 292: Change "on" with "in"

Authors response: done

Lines 316-317: Change with "Additionally, there is a growing concern towards ....."

Authors response: done

Line 320: change with "to the understanding"

Authors response: done